# Design and Experiment of an Automatic Row-Oriented Spraying System Based on Machine Vision for Early-Stage Maize Corps

Kang Zheng [1,2,†], Xueguan Zhao [3,†], Changjie Han [2], Yakai He [4], Changyuan Zhai [1,3,*] and Chunjiang Zhao [1,5,*]

1    Intelligent Equipment Research Center, Beijing Academy of Agriculture and Forestry Sciences, Beijing 100097, China
2    College of Mechanical and Electrical Engineering, Xinjiang Agricultural University, Urumqi 830052, China
3    National Engineering Research Center for Information Technology in Agriculture, Beijing 100097, China
4    Chinese Academy of Agricultural Mechanization Science Group Co., Ltd., Beijing 100083, China
5    College of Agriculture Engineering, Jiangsu University, Zhenjiang 530004, China
*    Correspondence: zhaicy@nercita.org.cn (C.Z.); zhaocj@nercita.org.cn (C.Z.);
     Tel.: +86-10-5150-3886 (C.Z. & C.Z.)
†    These authors contributed equally to this work.

**Abstract:** Spraying pesticides using row alignment in the maize seedling stage can effectively improve pesticide utilization and protect the ecological environment. Therefore, this study extracts a guidance line for maize crops using machine vision and develops an automatic row-oriented control system based on a high-clearance sprayer. First, the feature points of crop rows are extracted using a vertical projection method. Second, the candidate crop rows are obtained using a Hough transform, and two auxiliary line extraction methods for crop rows based on the slope feature outlier algorithm are proposed. Then, the guidance line of the crop rows is fitted using a tangent formula. To greatly improve the robustness of the vision algorithm, a Kalman filter is used to estimate and optimize the guidance line to obtain the guidance parameters. Finally, a visual row-oriented spraying platform based on autonomous navigation is built, and the row alignment accuracy and spraying performance are tested. The experimental results showed that, when autonomous navigation is turned on, the average algorithm time consumption of guidance line detection is 42 ms, the optimal recognition accuracy is 93.3%, the average deviation error of simulated crop rows is 3.2 cm and that of field crop rows is 4.36 cm. The test results meet the requirements of an automatic row-oriented control system, and it was found that the accuracy of row alignment decreased with increasing vehicle speed. The innovative spray performance test found that compared with the traditional spray, the inter-row pesticide savings were 20.4% and 11.4% overall, and the application performance was significantly improved.

**Keywords:** maize; machine vision; pesticide application; navigation tracking

## 1. Introduction

As the main food in the world, maize has made outstanding contributions to human survival, occupying a significant strategic position for food security [1]. In the early maize growth stage, farmers spray pesticides three–five times a year to treat pests, but this method leaves a large amount of pesticides deposited on the soil surface. Therefore, some key problems urgently need to be solved to protect planting ecology by reducing pesticide waste and improving pesticide utilization [2]. The lack of precision in pesticide application has aroused wide concern in agricultural science, so intelligent pesticide application equipment has become an important tool to improve efficiency [3–5]. Plant growth and canopy changes tend to misalign the spraying nozzle and the plant canopy when relying on manual operation for row alignment, resulting in repeated or missed spraying by the

sprayer [6,7]. Maize is a row drill crop, so visual sensors can be used to extract image information according to planting features, achieving precise pesticide application to rows of crops, improving the pesticide utilization rate and reducing pesticide residues [8–10]. The key technologies for realizing automatic row alignment based on machine vision are sensing and tracking.

In terms of sensing, machine vision has a low cost and large information carrying capacity, but the stability of row guidance in fields is often limited by the time-consuming and changeable outdoor environment [11–13]. Therefore, row guidance development should find ways to design a real-time crop row extraction algorithm with high identification accuracy and smooth operation [14–19]. In recent years, scholars have proposed many row guidance methods, including grayscale images, binary images, feature point extraction and crop row fitting.

In the process of row guidance extraction, because crop rows are not displayed in parallel in an image, it is difficult to identify crop rows. Therefore, on the basis of strip segmentation, the corresponding feature points can be obtained through vertical projection. Zhou et al. [20] used a horizontal stripe to determine the initialization midpoint, divided the region of interest with the initial midpoint as the center, and determined the feature points of crop lines through the vertical projection in the region of interest. This method can reduce the non-characteristic area of an image and improve operation efficiency. Ospina and Noguchi detected the crop centroid in a horizontal strip, found the geometric center of the crop rows, and fit the geometric center based on the least squares method to obtain a guidance line [21]. A Hough transform distinguishes image features by geometric shapes and has become a common method for crop row identification. It is beneficial for avoiding weed noise, but the method takes a long time to process. Rovira-Más uses a Hough transform to extract crop rows in the region of interest. To improve the efficiency and quality of image processing, Rovira-Más adjusted the appropriate threshold range, set up target points in the region of interest, and found the best path using connectivity analysis [22]. Chen proposed a crop row fitting method based on an automatic Hough transform accumulated threshold. The parameter points on the accumulated plane were clustered using K-means, and the best accumulated threshold was obtained based on the difference in distance of the cluster centroid and the variance within the group. The centroid of the accumulated plane cluster under the best accumulated threshold was used as the fitting line of the crop rows. This method had a high identification success rate, but the clustering algorithm increased the overall time consumption [23].

For tracking, Bakker designed an automatic navigation system based on a real-time differential global positioning system (RTK-DGPS) and developed a row guidance method combining RTK-DGPS with machine vision, achieving path tracking using a PID control algorithm [24]. Based on its Robocrop system, Garford Company adjusts the lateral hoe offset using a hydraulic side shift, and realizes interrow tracking [25]. The American company John Deere designed the AutoTrac sprayer navigation system [26] by adopting a sensing scheme combining machine vision with mechanical sensors to address the seeder navigation system drift and "wavy" crops in the artificial planting field problems. Zhang used LiDAR and a new type of mechanical alignment sensor to build a crop row sensing system for maize harvesters and designed an automatic row alignment control system for maize harvesters based on fuzzy PID control and a tracking model. The average deviations of a lateral offset test using LiDAR and a multisensor test in the field were 0.0775 m and 0.146 m, respectively, and the average deviation of the mechanical sensor test was 0.0876 m [27]. Although hydraulic control has a fast response speed, the control process is complicated and difficult to realize in practice. Therefore, an urgent engineering problem that needs to be solved is to select an appropriate control scheme and incorporate a sensing scheme to realize the integrated application of the system and improve the alignment accuracy.

To carry out row guidance algorithms, most existing studies focus on images, but there are few reports of research on the use of video reasoning algorithms for improving the real-

time performance and robustness of sensing and integrated development of control systems. This study aims to design a novel row guidance extraction algorithm, stressing solving the robustness problem affected by the external environment in real-time extraction and video processing to obtain the guidance information of the tracking path. Meanwhile, a visual automatic row-oriented spraying system was developed based on autonomous navigation for a high-clearance sprayer. After a camera obtained the guidance line information in the detection area, a row-oriented delay compensation model was established. By controlling the slide boom, the lateral movement of the spray nozzles was accurately adjusted to achieve crop-row tracking and spraying to improve the pesticide utilization rate.

## 2. Materials and Methods

### 2.1. System Description

#### 2.1.1. System Composition

To achieve precise localization of the spraying system between crop rows, a visual row-oriented spraying platform with autonomous navigation was built based on a high-clearance sprayer. As shown in Figure 1, the test platform consists of a navigation system, vision system, row alignment control system, sliding boomer, application supply system and travel chassis. The antennas of the navigation system (AMG-1202, National Agricultural Intelligent Equipment Engineering Technology Research Center, Beijing, China) are symmetrically installed on both sides of the test platform. The vision system uses a network camera (LPCP10190_1080P, Xinjiahua Electronics Co., Ltd., Shenzhen, China) with an image resolution of 1920 × 1080 pixels, a focal length of 2 mm and a monitoring angle of 90°. The camera is installed at the center of the front of the test platform, and the installation angle is 60° to the ground. The key component of the row alignment control system is an electric linear actuator (TG-300B, Shenzhen Bosgoal Technology Co., Ltd., Shenzhen, China), and the maximum stroke of the electric actuator corresponds to the plant row spacing. The width of the sliding boomer is 12.2 m, providing 22 spraying locations, and the maximum height from the ground is 1.4 m. A three-cylinder plunger pump was selected for the application supply system, with a liquid pump flow rate of 36~81 L/min and a spraying pressure range of 0.05~0.6 MPa. The travel chassis uses a four-wheel drive, with a wheel track of 1.6 m and a chassis height of 1.2 m, ensuring the stable operation of the platform in the high-clearance spraying environment.

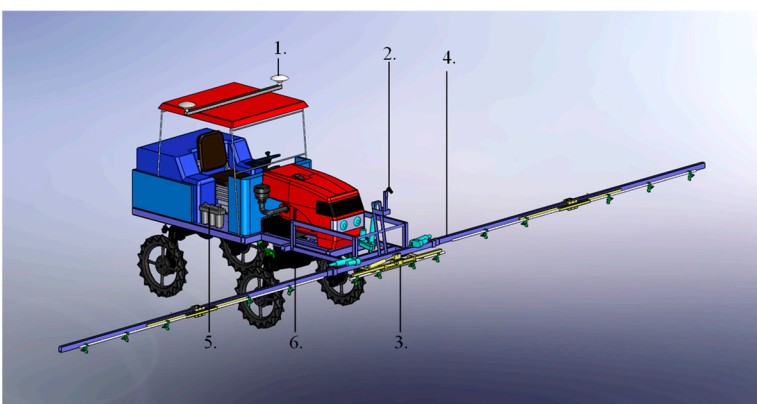

**Figure 1.** Structure of the visual row-oriented spraying platform: 1. Navigation system 2. Vision system 3. Row-oriented control system 4. Sliding boomer 5. Application supply system 6. Travel chassis.

#### 2.1.2. Principle of the Visual Row-oriented Spraying System

A high-clearance sprayer equipped with an autonomous navigation system can preliminarily align the drill plants [28–30]. The visual row-oriented principle is shown in Figure 2. The camera transmits the collected images to the computer, and then the computer calculates the angle information and offset information of the crop rows through the crop row identification algorithm and sends them to the electronic control unit (ECU). The speed

encoder of the high clearance sprayer provides the traveling speed v of the test platform, and the ECU analyzes the delay control time t according to the row-oriented delay model, and triggers the control signal. The motor driver applies pulse-width modulation (PWM) to control the reciprocating movement of the sliding boom, and the Hall encoder on the electric linear actuator obtains the alignment offset in real time and feeds it back to the ECU, thereby adjusting the lateral movement of the nozzle and achieving closed-loop control of the sliding boom.

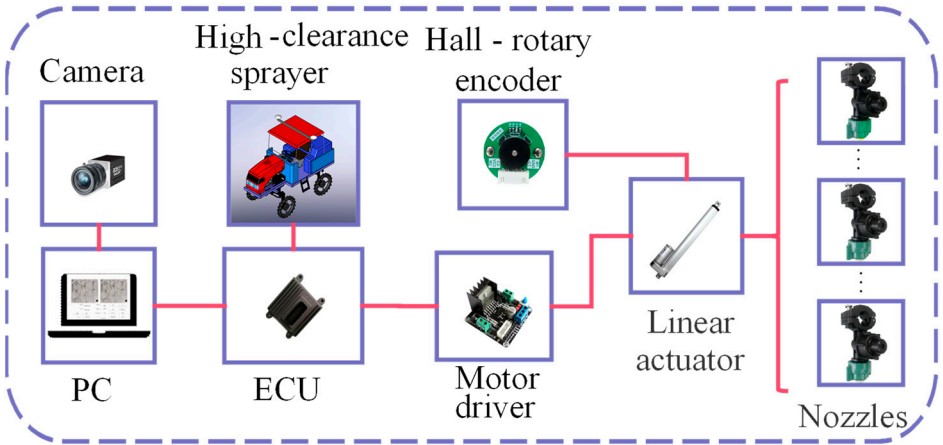

**Figure 2.** Composition of the visual row-oriented spraying system.

## 2.2. Visual Algorithm Design

### 2.2.1. Development Environment and Technical Process

The video acquisition equipment is a GoPro Hero9 (GoPro Inc., SAN Mateo, California, USA), which is 1.2 m above the ground, with an image resolution of 1920 × 1080 pixels and video frame rate of 30 fps. C++ is used as the programming language, and the OpenCV open-source library was selected for algorithm realization. The compiler is Visual Studio 2019, the image processing hardware is a PC, the processor is an Intel(R) Core i7-1065G7, the graphics card is an NVIDIA GeForce MX350, and the memory is 16 GB.

The guidance line extraction process is shown in Figure 3. After the camera collects an image of the crop rows, it uses a segmentation algorithm to extract the target and morphological operations and extracts the region of interest (ROI) of the image. Based on a Hough transform, the candidate crop rows were obtained in the polar coordinate space. Outlier lines were filtered in the candidate crop rows through straight line slope features, thus obtaining auxiliary lines. The guidance line was extracted according to the tangent formula of the included angle. If the guidance angle is greater than 5° and the lateral offset Δ is greater than 20 cm, it is regarded as information distortion, and the previous guidance information frame is called for information compensation. If the judgment condition is present, the guidance line is filtered using a Kalman filter, the guidance line information is output, and the next frame of the image is circularly updated. Guidance line information was calculated using a row alignment control algorithm, thus achieving visual row-oriented spraying.

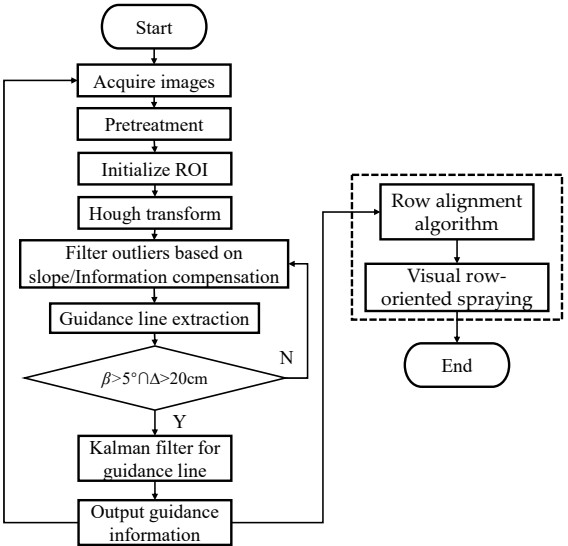

**Figure 3.** Guidance line extraction procedure for row-oriented spraying.

### 2.2.2. Image Pretreatment

Image pretreatment is the key step of crop row extraction in a field, aiming to completely segment the plants from the soil and obtain the characteristic plant information. To explore the effect of changing light on color channels, 100 groups of R, G and B color channel pixel values for maize and soil were measured under different light conditions. The statistical results in Figure 4a show that the brightness of the soil B channel was higher than that of the R and G channels, and the brightness of the maize G channel was much higher than that of the R and B channels. Therefore, the segmentation algorithm adopted the coupled G-R and G-B channel characteristics, thus improving the traditional Excess green algorithm, and the maize image in Figure 4b was converted to grayscale. To reduce the influence of illumination, the values of the R, G and B channels were normalized, and the processed results were replaced using the existing color channels r, g and b. The mathematical expression is shown in formula (1). The grayscale image is shown in Figure 5c.

$$
\begin{cases}
r = \frac{R}{R+G+B} \\
g = \frac{G}{R+G+B} \\
b = \frac{B}{R+G+B} \\
r + g + b = 1
\end{cases}
\tag{1}
$$

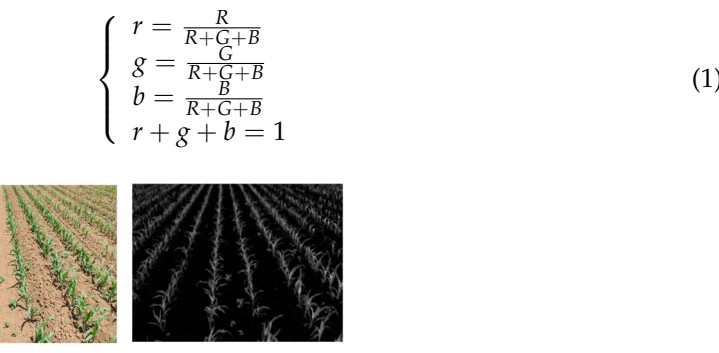

| (**a**) | (**b**) | (**c**) |

**Figure 4.** (**a**) Pixel histogram; (**b**) Original image; (**c**) Grayscale image.

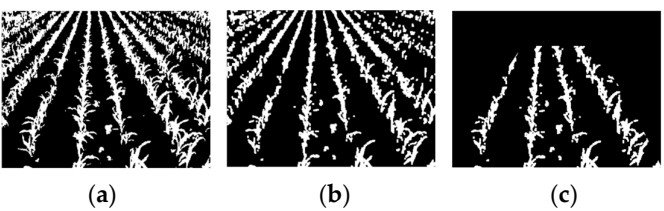

| (**a**) | (**b**) | (**c**) |

**Figure 5.** (**a**) Binary image; (**b**) Morphology image; (**c**) ROI image.

After the grayscale image was obtained, Otsu's method was applied to obtain the dynamic threshold, thus obtaining a binary image, as shown in Figure 5a. In addition, the binary image was morphologically processed to eliminate noise and reduce holes, and the processing results are shown in Figure 5b. When working, the machine only needs to pass through the specified path, but there was considerable interference information in the images collected by the camera. Therefore, it was necessary to carry out ROI processing on the images to reduce the data redundancy of the algorithm and improve operation efficiency. The processing results are shown in Figure 5c.

### 2.2.3. Crop Row Identification

Because the camera shoots at an inclination angle, the crop row images obtained were not parallel to each other, so it was necessary to use vertical projection to divide the horizontal strip in the ROI to obtain feature points. It is known that the image size is w × h, the width is w, the height is h, the strip size in the ROI is w × Δh, and the number of strips in the ROI is m = h/Δh. To obtain enough feature points, m was set to 8. The independent contours in each strip were traversed from left to right and the geometric centroid of all contours was found as the feature point coordinates [31]. One strip was regarded as a group, and feature points of the strip were stored in the queue. The visualization results of the features are shown in Figure 6.

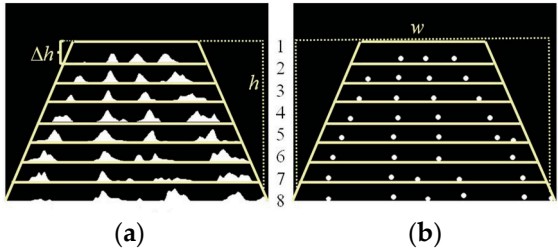

(a)　　　　　　　　　　　(b)

**Figure 6.** (**a**) Vertical projection; (**b**) Feature point collection.

As shown in Figure 7, the Hough transformation mapped the linear parameters in the rectangular coordinate system to the parameter space in the polar coordinate system, converted the collected feature points into curves in polar coordinates, and determined the peak points of the fitted straight lines from the number of intersections of curves [32]. The parameter equation of a straight line in polar coordinates is $\rho = x\cos\theta + y\sin\theta$, where $\rho$ represents the vertical distance from the straight line to the origin, and $\theta$ represents the angle from the *x*-axis to the vertical of the straight line, with a value range of $\pm90°$. Therefore, the linear equation in the rectangular coordinate system can be transformed into parameter space coordinates $(\rho,\theta)$.

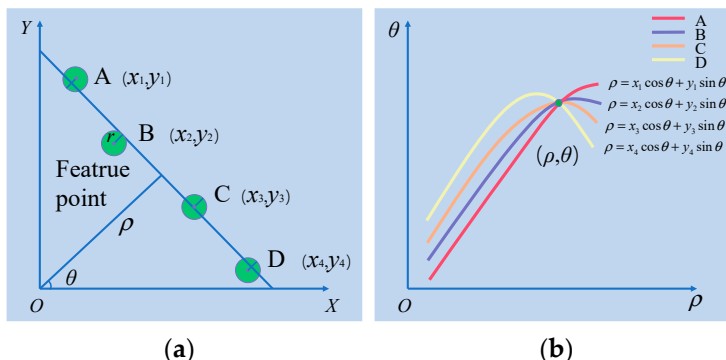

(a)　　　　　　　　　　　(b)

**Figure 7.** (**a**) Rectangular coordinate system; (**b**) Polar coordinate system.

Points A, B, C and D in Figure 7a are the feature points of the feature point set, which are not strictly linearly distributed. Therefore, each feature point was taken as the center, the

neighborhood radius R of the feature point was set, and all the pixels in the neighborhood of the feature point were mapped to the parameter space. As shown in Figure 7b, the Hough transform accumulated the pixels with the same ($\rho$,$\theta$) parameters in feature points A, B, C and D, and judged the straight lines in the feature points that meet the fitting conditions using the number of accumulated values of the ($\rho$,$\theta$) parameters. Therefore, it was necessary to set the polar diameter $\rho$, polar angle $\theta$, and accumulated threshold thr in the parameter space to fit the straight line and set the polar diameter resolution $\Delta\rho$ to 1 pixel and the polar angle resolution $\Delta\theta$ to 1 rad. By adjusting the accumulated threshold thr, the optimal Hough transform for straight line fitting in the image could be obtained, and the processing result is shown in Figure 8a.

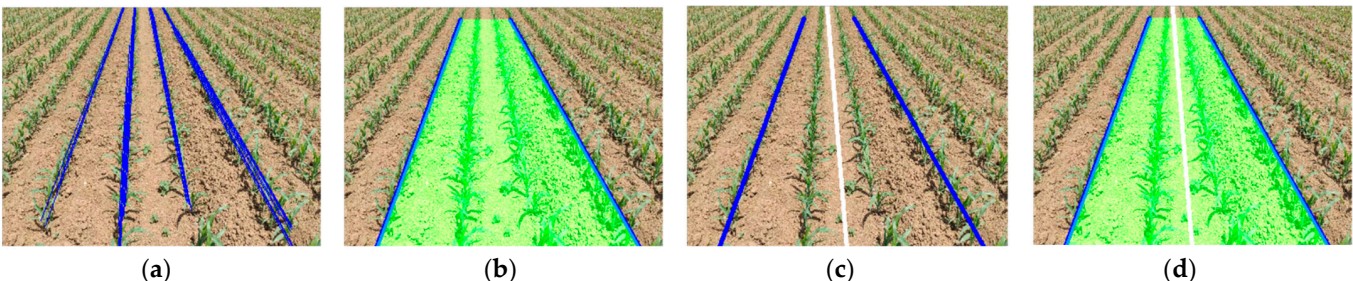

| (a) | (b) | (c) | (d) |

**Figure 8.** (**a**) Hough transform; (**b**) auxiliary lines; (**c**) guidance line; (**d**) final result.

### 2.2.4. Guidance Line Extraction and Tracking

Candidate crop rows can be obtained with the Hough transform, but many crop rows cannot provide specific row guidance information to the control system. Therefore, the straight-line slope features were used to filter outliers, eliminate invalid data, and obtain two auxiliary lines under the tracking path.

As shown in Figure 8a, there were many line clusters on the crop rows that can be classified by the positive and negative slopes of straight lines, and the left and right sets of straight-line subsets $I_l$ and $I_r$ can then be obtained. Outlier filtering was carried out on subsets $I_l$ and $I_r$. First, the subsets were averaged; second, the absolute value of subset element $O_i$ was obtained as the difference with the mean $O_{mean}$, and the new value $S_i$ was stored in the new subset $I_s$; third, the subset $I_s$ was traversed to find the maximum value $S_{max}$, if $S_{max}$ was greater than the outlier threshold $T_j$, the elements indexed by $S_{max}$ in $I_s$ and I were removed until the subset met the outlier threshold condition. Finally, the filtered subset elements were returned. Among them, the outlier threshold $T_j$ was set to 0.2 based on the principle of preferring small, and the pseudo code of Algorithm 1 is as follows:

In summary, the algorithm was able to filter outliers with large errors according to the slope features and obtain two auxiliary lines on the left and right sides of the test platform driving path. The processing results are shown in Figure 8b.

To extract the test platform traveling path guidance line, the left and right auxiliary lines of the crops were taken as the reference, in which the slopes of the auxiliary lines of the left and right crops were recorded as $\eta_1$ and $\eta_2$. Using Formula (2) to obtain the guidance line slope $\eta$ of the traveling path, the result is shown in Figure 8c.

$$\frac{|\eta - \eta_1|}{|1 + \eta\eta_1|} = \frac{|\eta - \eta_2|}{|1 + \eta\eta_2|} \tag{2}$$

The test platform is in an unstructured environment where various random factors exist that may affect the visual data collected by the camera, such as the vibration of the chassis, the disturbance of crop leaves with the wind, and the sudden change in illumination. As a result, when the computer is processing continuous video frames, the guidance line often appears to jitter. To solve this problem, the navigation line was taken as

the prediction object and a Kalman filter was used to optimize the estimation and determine the guidance information.

$$\begin{cases} x_k = A_k x_k + B_k u_k + w_k \\ z_k = H_k x_k + v_k \end{cases} \tag{3}$$

---

**Algorithm 1.** Slope-based outlier filter.

---

**Input**: a set of objects I = {$O_1$, $O_2$, $O_3$ ... $O_n$}, threshold = $T_j$, Is = $\varnothing$
**Output**: inliers in I
**Method**:
**Initialize**: I = $I_l$ or $I_r$, $T_j$ = 0.2, $O_{sum}$ = 0, count = 0
**while**
**for** I = 1 **to** n **do**
    $O_{sum}$ = $O_{sum}$ + $O_i$
    count = count + 1
**endfor**
$O_{mean}$ = $O_{sum}$/count
**for** i = 1 **to** n **do**
$S_i$ = $O_i$ − $O_{mean}$
**Push** $S_i$ **into** $I_s$
**endfor**
**Find** $S_{max}$ **in** $I_s$
**if** $S_{max}$ > $T_j$ **then**
i = **idx**($S_{max = i}$)
**pop** $S_i$ **from** $I_s$
**pop** $O_i$ **from** $I_s$
    **else**
     **break**
    **endif**
**return** I

---

In the unstructured environment, with a discrete extraction process, a set of state space equations should be introduced to describe the Kalman filter model. Formula (3) is the state equation, $x_k$ is the system state of the guidance line at time $k$, $A_k$ is the state transition matrix, $u_k$ is the system control quantity at time $k$, $B_k$ is the control matrix and $w_k$ is the system error. Formula (3) is the measurement equation, $z_k$ is the observed value at time k, $H_k$ is the measurement matrix and $v_k$ is the measurement error.

Second, assuming that the current state of the system is $k$, using the system process model to predict the current state according to the previous state of the system, the state update model Formula (4) can be obtained.

$$\begin{cases} x_{k|k-1} = A_k x_{k|k-1} + B_k u_k \\ P_{k|k-1} = A_k P_{k|k-1k} A_k^T + Q_k \\ K_k = P_{k|k-1} H_k^T (H_k P_{k|k-1} H_k^T + R_k)^{-1} \\ x_{k|k} = x_{k|k-1} + K_k (z_k - H_k x_{k|k-1}) \\ z_{k|k} = (1 - K_k H_k) P_{k|k-1} \end{cases} \tag{4}$$

where $x_{k|k-1}$ is the predicted state value at $k-1$, $B_{k|k-1}$ is the predicted minimum mean square error at $k-1$, $K_k$ is the Kalman gain, $x_k$ is the corrected state value and $z_{k|k}$ is the corrected minimum mean square error. During the test, the extraction error caused by camera shaking was the largest, which was regarded as the measurement error. The measurement noise variance matrix $R_k$ was 0.1, the plant leaves "perturbing" with the wind was the smallest error and was process noise, the system noise variance matrix $Q_k$ was 0.001, and the initial minimum mean square err or $P_{k|k-1}$ was 0.1. The final result is shown in Figure 8d.

### 2.3. Row Alignment Control Method

2.3.1. Visual Localization

Visual row alignment control was carried out as follows. After obtaining the guidance information, the image coordinates of the guidance line aiming point were converted into the actual coordinates in the world coordinate system to provide guidance information to the control system. According to Figure 9a, the camera was arranged above the crop canopy at a fixed height and shot at an inclination angle to the ground. $H$ is the height of the camera from the top of the canopy, $D_{min}$ is the actual distance between the bottom edge of the image and the camera, $D_{max}$ is the actual distance between the top edge of the image and the camera, $\alpha$ is the camera pitch angle and $\theta$ is the vertical angle of the field of view. According to the geometric relationship of the above parameters, the equivalent relation Formula (5) is presented. $\Delta\theta$ and $\theta$ were linearly related to the actual coordinates in pixel coordinates, so the actual ordinate $y_1$ of the aiming point was obtained according to Formula (5).

$$\begin{cases} \alpha = \arctan\frac{D_{min}}{H} \\ \theta = \arctan\frac{D_{max}}{H} - \alpha \\ \Delta\theta = \frac{(height - y_0)}{height}\theta \\ y_1 = H \bullet \tan(\alpha + \Delta\theta) \end{cases} \tag{5}$$

$W$ is the maximum distance of the field of view and the horizontal field angle, and the corresponding relation Formula (6) was established from Figure 9b to obtain the actual abscissa $y_0$ of the aiming point.

$$\begin{cases} \frac{W}{2} = D_{max} \bullet \tan\frac{\beta}{2} \\ x_1 = \frac{W \bullet (x_0 - width/2)}{width} \end{cases} \tag{6}$$

where *height* is the image pixel height, *width* is the image pixel width, $(x_0, y_0)$ is the pixel coordinate of the aiming point, and $(x_1, y_1)$ is the actual coordinate of the aiming point.

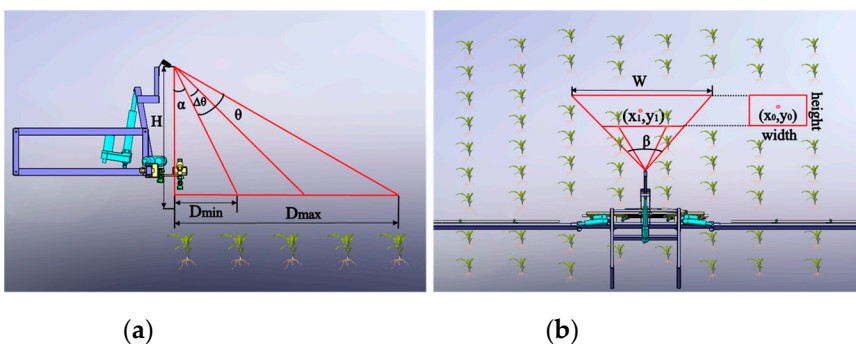

|           |           |
|:---------:|:---------:|
| (**a**)   | (**b**)   |

**Figure 9.** Visual localization.

2.3.2. Row-Oriented Delay Compensation Model

Precise row alignment control requires that the system incorporate guidance information to establish a row-oriented delay compensation model. As shown in Figure 10, area A is the image field of view, area B is the blind area of the field of view, the guidance line is $y = \eta x + c$, and the lateral distance $l_2$ of the nozzle is the relative abscissa $x_1$ from the aiming point P to the absolute coordinate system $O$ on the guidance line. As shown in Figure 11, the camera was installed at the centerline of the front end of the test platform, and the spraying nozzle was installed on the sliding boom. There was a distance between the aiming point of the camera and the spraying nozzle in the vertical direction, that is, the longitudinal distance $l_1$ between point P and the spraying nozzle. Therefore, it can be inferred that when the linear actuator receives the execution signal, it cannot execute immediately, so it needs time delay control. First, the execution time of the system components was theoretically analyzed, and the theoretical delay compensation time theory of the row-oriented control system was obtained. The compensation time includes identification and localization time

$t_1$, communication time $t_2$, and execution time $t_3$ for the bank mechanism, wherein the identification and localization time $t_1$ was 32 ms on average after repeated tests using the *GetTickCount*() function. In addition, the visual row-oriented control communication mode used USB-to-CAN packet mode, with a serial port baud rate of 115,200, a fixed packet length of 16 bytes and a communication time $t_2$ of 1 ms [33]. At the same time, the stroke of the linear actuator was 500 mm, the theoretical speed was 240 mm/s, and the maximum theoretical execution time $t3$ was 2.08 s. To ensure the motion stability of the linear actuator, it was necessary to set a speed safety boundary for the delay compensation time model according to the theory. Only when the time taken to advance the implement $l_1$ meets the speed safety boundary condition can the sliding boom perform the completion offset D. Therefore, a delay compensation model of the row-oriented mechanism was established, and the speed obtained by the encoder on the test platform is $v$, so that the delay execution time $t$ of the linear actuator can be obtained according to Formula (7).

$$\begin{cases} t_{theory} = t_1 + t_2 + t_3 \\ v < v_{theory} \leq \dfrac{l_1}{t_{theory}} \\ t = \dfrac{l_1}{v} \end{cases} \tag{7}$$

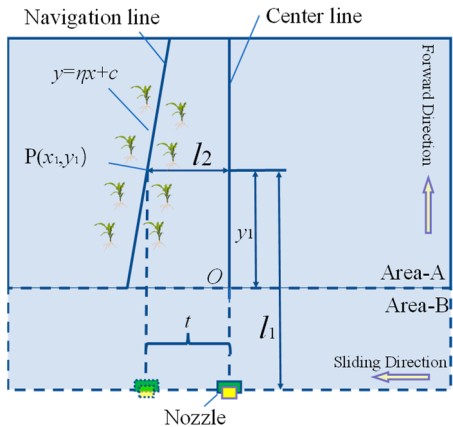

**Figure 10.** Delay compensation model.

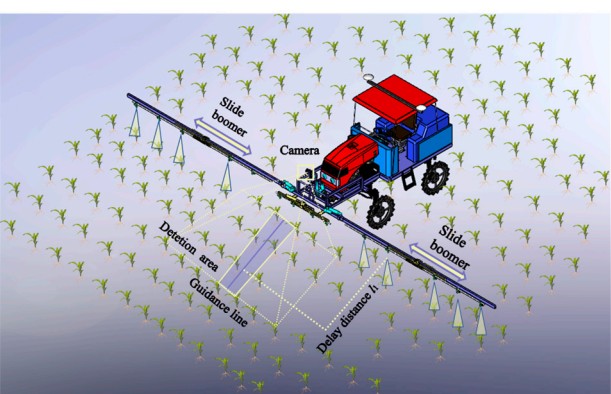

**Figure 11.** Visual row-oriented spraying platform in maize fields.

### 2.3.3. Row-Orientation Control Algorithm

To reduce the frequent start and stop of the DC motor and increase its service life, PWM full-speed regulation was adopted. The technical flow chart of the control system is shown in Figure 12. First, the system is initialized, the ECU triggers the timer, and the sliding boom moves to the middle of the stroke so that the boom is aligned with the center of the vehicle. Second, the ECU circularly receives the guidance information, analyzes the serial port data, and enters the delay control through the above row-oriented delay

compensation model. The electric linear actuator encoder provides feedback on the position of the sliding boom in real time, and the ECU requests and calculates the difference between the target position and the previous position of the sliding boom. When the calculation result is less than the target position, the sliding boom shifts to the left, and when it is greater than the target position, the sliding boom shifts to the right. If the sliding boom reaches the target position, the motor stops moving, and the control ends.

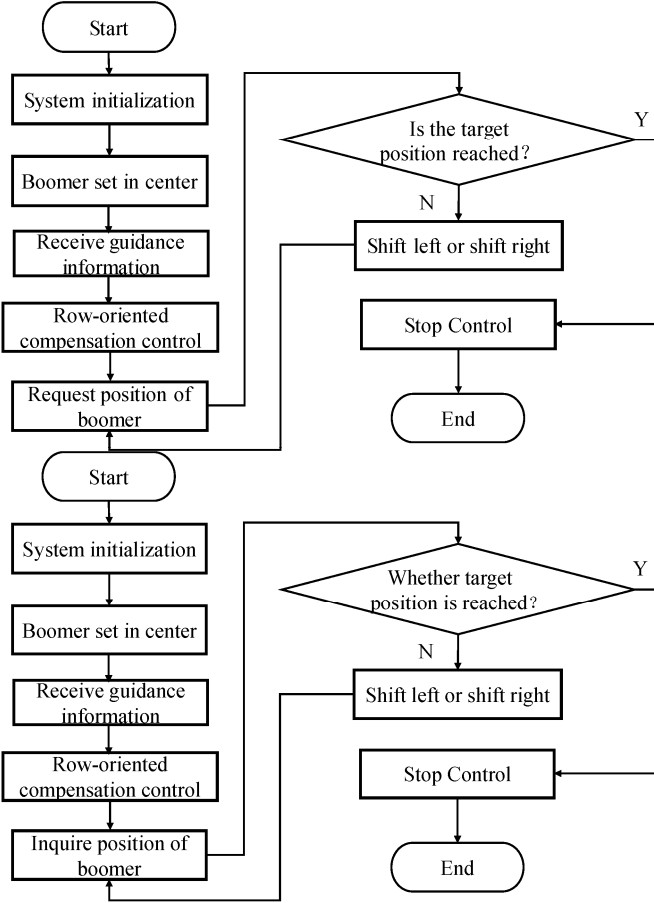

**Figure 12.** Flow chart of the row-orientation control algorithm.

*2.4. Test Method*

2.4.1. Robustness Test of Guidance Line Algorithm

Considering the adaptability of the algorithm in the field, all-day images of maize seedlings were collected at the Xiaotangshan National Precision Agriculture Research Demonstration Base, and the illumination intensities ranged from 4.57~6.09 wLux, 7.43~8.54 wLux, and 10.27~11.36 wLux, respectively. The growth period of maize has three stages: (average growth height is 30 cm), early jointing stage (average growth height is 40 cm) and middle jointing stage (average growth height is 80 cm). To verify the robustness of the algorithm in an unstructured environment, 180 video frames were tested. When the algorithm can extract two auxiliary lines of crop lines and fit the guidance line, the identification is considered successful.

2.4.2. Guidance Line Algorithm Accuracy Test

In the process of algorithm design, the accumulated threshold of the Hough transform is the key factor that affects the accuracy of the guidance line, so the accumulated threshold of the Hough transform was optimized. In each group, 60 video frames were randomly selected. Based on a manually extracted guidance line [16], under different accumulated thresholds, the guidance angle and offset were measured. When the error between the guidance angle processed by the algorithm and the angle extracted manually is less than

5° and the offset error between them is less than 5 cm, identification is considered to be accurate.

### 2.4.3. Kalman Filter Optimization Test

In the process of video reasoning, the guidance angle change is not obvious, but its lateral offset fluctuates greatly [9,34], so guidance offset was introduced as an experimental index. To test the inhibition effect of the Kalman filter on random factors in the field, 50 consecutive video frames were taken from the collected video of crop rows, and the angle and offset of the guidance line were extracted using the guidance line identification algorithm. The original information in the video frame without the Kalman filter and the filtered information were printed at the same time for a comparison test. The video capture showed that the vehicle speed was 0.5 m/s, the wind speed was 2.89 m/s, the light intensity was 7.43 wLux and the temperature was 36.2 °C.

### 2.4.4. Control System Test

A row alignment control system was designed according to a row-orientation delay compensation model. To test the row-orientation performance of the system, a test was conducted in the threshing ground at the Xiaotangshan National Precision Agriculture Research Demonstration Base. According to the planting requirements, two rows of plants were arranged manually, and a position coordinate ruler was placed between the rows of plants. To replicate an actual situation that may occur in the field, three rows of plants were designed. The first was parallel to the vehicle's traveling direction, the second was inclined to the vehicle's traveling direction, and the third was "S"-shaped. To record the tracking path of the sliding boom, the funnel was fixed at the centerline of the spraying boom. The test platform runs at a constant speed of 0.5 m/s, and the sliding boom acts according to the visual signal. The funnel leaves sand tracks on the ground along with the sliding boom. The lateral deviation between plants and sand tracks was manually measured, and the position coordinates were recorded.

### 2.4.5. Field Test

To verify the visual row-oriented effect of autonomous navigation, a field test was conducted in the maize field of the Xiaotangshan National Precision Agriculture Research Demonstration Base, as shown in Figure 13. During the test, the track of the sliding boom was recorded, and after the test, it was compared with the corresponding plant position to evaluate the row-orientation accuracy. Because it is necessary to introduce a comparison with and without autonomous navigation, the A-B autonomous navigation point trajectory was set in advance. Before the test, a section of 20 m was marked as the test area, and 5 m was reserved in front of each section of the test area as a test platform acceleration and deceleration buffer. A handheld RTK was used to record the plant position in each section of the test area, which was used as the baseline of the row alignment effect. During the test, the RTK was fixed on the centerline of the sliding boom to record its movement track. Three test speeds of high, medium and low were designed, and at each speed, the test control groups were set at the state of on or off navigation. A total of six groups of tests were conducted.

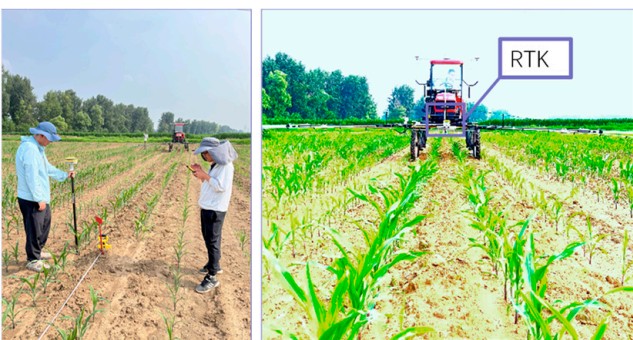

**Figure 13.** Field test layout.

2.4.6. Droplet Deposition Test

A comparative test between full-coverage spraying and row-oriented spraying was designed to explore the droplet deposition of the two conditions. According to the planting requirements, two rows of plants were arranged in the field, with 20 maize plants in each row, and 98 pieces of round fiber filter with a diameter of 9 cm were arranged between rows and plants. As shown in Figure 14, the filter paper center of mass was used as the sampling center, and a square of 30 cm × 30 cm was taken as the sampling area for droplet deposition. The spraying deposition amount in the sampling area is the product of the deposition amount per unit area on each filter paper and the sampling area. The site layout is shown in Figure 18. To reduce the droplet deposition between crops, a 110° fan nozzle was selected for full-coverage spraying, and a 40° fan nozzle was selected for row-oriented spraying. The test process is as follows: start the spraying system, set the spraying pressure at 0.27 MPa, then drive the sprayer at a speed of 0.5 m/s at a constant speed along the crop row direction. After the spraying operation, the fiber filters were quickly collected into a washing dish for droplet deposition determination.

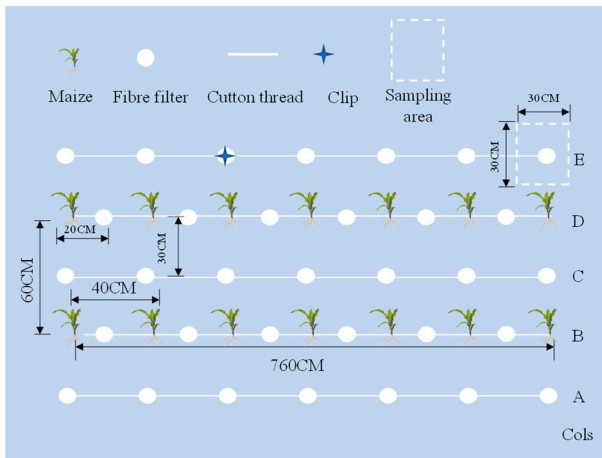

**Figure 14.** Sampling point layout.

As shown in Figure 15, the "fluorescence tracer technique" was selected for the droplet deposition test, and 0.134 g Rhodamine powder was added to 1 L of distilled water to prepare the Rhodamine solution. The Rhodamine solution was fully mixed with 200 L water in the tank of the sprayer, and the Rhodamine solution was elevated three times. According to the absorbance value of the tracer analyzer, the Rhodamine solution concentration was measured, and the average concentration was found to be 996.34 μg/L. After the spraying test, the sampling filter paper was washed at a constant volume, and the constant volume solution was 50 mL. The concentration of the constant volume solution after washing was measured using a tracer analyzer.

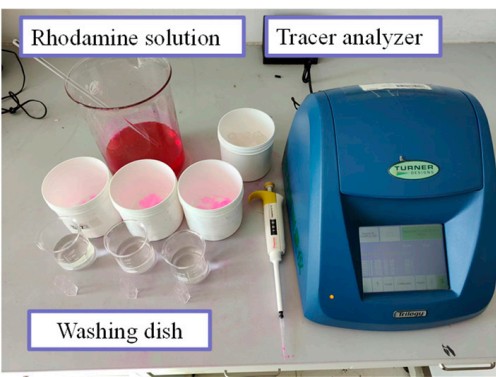

**Figure 15.** Test instrument.

## 3. Results

### 3.1. Robustness Test Results and Analysis

As shown in Figure 16, the leaf area, shape and density of plants in different growth periods were different. As shown in Table 1, the proportion of target pixels corresponding to the growth height was calculated, showing that it increases with the growth period. Moreover, when the maize grows to 30 cm and 40 cm, the identification success rate of the algorithm is higher, but when it grows to 80 cm, the identification success rate is difficult to guarantee. Because the leaf spacing of maize is clear and complete at heights of 30 cm and 40 cm, it can be clearly distinguished between plants and soil. At a height of 80 cm, the leaves were obviously enlarged, and overlapped with each other, close to "ridge sealing", increasing the difficulty of the morphological operation and thus affecting the identification success rate. Because the image acquisition period is in the dry season, the weed pressure is low, and the weeds in the field are mostly Inula and Portulaca oleracea, with an average weed density of 34 plants·m$^{-2}$. The algorithm uses the Hough transform to extract crop rows and reduces the influence of weed noise, so the identification success rate is not affected by weed density. The test images were collected all day, and the coverage of light intensity was 4.57~11.36 wLux, basically meeting the needs of agricultural production in various periods. As shown in Table 1, when the illumination range is too large, the identification accuracy decreases. Excessive exposure will increase the preprocessed target pixels, and the obtained feature points will be inaccurate, thus affecting crop row extraction. In addition, the algorithm running time for 180 experimental images was less than 44 ms, meeting the real-time extraction requirements. The tests of different growth periods and light intensities of maize showed that the identification success rate of the algorithm was above 90.0% when the growth period of maize was in the third stage (30 cm) and the early jointing stage (40 cm), and the light intensity ranged between 4.57~11.36 wLux. This result indicated that the identification algorithm had good robustness and real-time performance and basically met the field operation requirements of the whole maize seedling period.

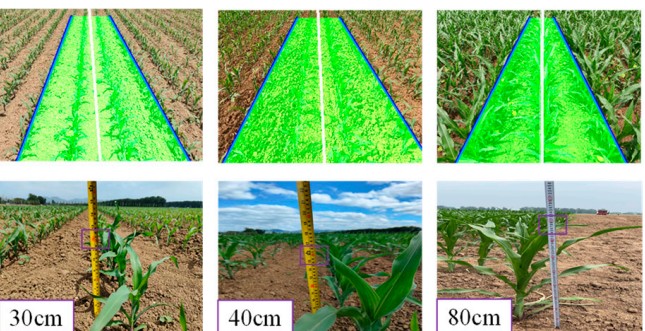

**Figure 16.** Test instrument.

**Table 1.** Robustness test.

| Image No. | Growth Height/cm | Illumination Intensity/wLux | Weed Density/(Plant·m⁻²) | Target Pixel Ratio/% | Algorithm Time/ms | Recognition Success Rate/% |
|---|---|---|---|---|---|---|
| 1~20 | | 4.57~6.09 wLux | 13~25 | 5.20–9.46 | 36–40 | 95 |
| 21~40 | 30 | 7.43~8.54 wLux | 14~28 | 5.5–8.18 | 34–39 | 100 |
| 41~60 | | 10.27~11.36 wLux | 14~28 | 6.89–9.57 | 33–44 | 90 |
| 61~80 | | 4.57~6.09 wLux | 11~26 | 6.7–8.2 | 34.5–41.3 | 100 |
| 81~100 | 40 | 7.43~8.54 wLux | 11~29 | 8.5–11.8 | 35.5–38.9 | 100 |
| 101~120 | | 10.27~11.36 wLux | 11~30 | 7.0–12.9 | 35.1–40.1 | 95 |
| 121~140 | | 4.57~6.09 wLux | 14~44 | 9.1–13.8 | 30.9–36.4 | 85 |
| 141~160 | 80 | 7.43~8.54 wLux | 15~39 | 12–15.5 | 31.3–35.7 | 90 |
| 161~180 | | 10.27~11.36 wLux | 17~42 | 11.6–16.3 | 32.5–36.1 | 80 |

*3.2. Accumulated Threshold Influence on Guidance Line Accuracy*

As shown in Figure 17, when the accumulated threshold thr is 25, the candidate crop rows are redundant, and too many lines will increase the weight of the outlier threshold, affecting the accuracy of the guidance line. When the accumulated threshold thr is 45, some candidate crop rows will disappear, affecting the guidance line fitting. When the accumulated threshold thr is 35, the number of extracted crop rows is relatively balanced, demonstrating an ideal fitting effect.

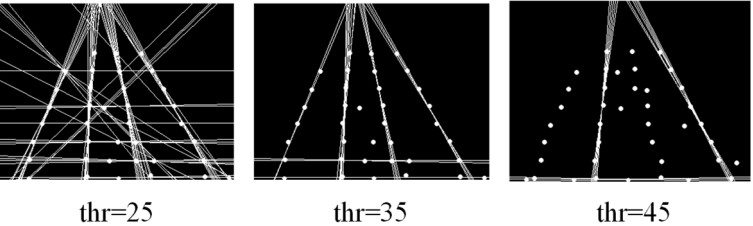

thr=25  thr=35  thr=45

**Figure 17.** Hough transform at different accumulated thresholds.

The test results shown in Table 2 indicate that when the accumulated threshold thr is 35, the average angle error, average offset error and identification accuracy are the best, which corresponds to the above analysis of the accumulated threshold. From the influence of the accumulated threshold on the angle error, the average angle error of the three groups of tests changed to different degrees, and the standard deviation of the angle error of the first group was 2.7° because of the redundancy of candidate crop rows. Compared with test group 2, the dispersion degree of angle error samples in group 1 is higher, and the angle error is larger. When the accumulated threshold is 45, the angle error fluctuates more. The observed offset errors of the three test groups show that the average values of the offset errors of the three groups are obviously different, which are 3.8 cm, 1.8 cm and 4.2 cm. Therefore, the accumulated threshold has a great influence on the accuracy of the guidance line. In addition, the standard deviation of the offset error of group 1 and group 3 is large, so it can be inferred that an inappropriate accumulated threshold will cause missing or redundant candidate crop rows and then affect the identification accuracy. When the accumulated threshold thr is 35, the best identification accuracy rate of this study is 93.3%, close to the data presented by Jiang et al. [35]. In addition, this article introduces lateral offset as a test index, which will provide a more reasonable reference for identification accuracy research.

**Table 2.** Accuracy test.

| Test Group | Hough Transform Accumulated Threshold | Angle Error Standard Deviation/° | Average Angle Error/° | Offset Error Standard Deviation/cm | Mean Offset Error/cm | Identification Accuracy/% |
|---|---|---|---|---|---|---|
| 1 | 25 | 2.45 | 2.7 | 3.8 | 2.6 | 88.3 |
| 2 | 35 | 0.74 | 0.9 | 1.8 | 1.1 | 93.3 |
| 3 | 45 | 2.80 | 3.6 | 4.2 | 3.2 | 86.0 |

*3.3. Evaluation of the Kalman Filtering Effect*

As shown in Figure 18, the original guidance information is frequently impacted by the vibration of the vehicle and the disturbance of leaves, appearing as serious jitter in the video stream. Moreover, in this case, the guidance signal cannot be transmitted to the control system. The Kalman filter data shows that the fluctuation of the heading angle and lateral offset of the guidance line processed by the Kalman filter is significantly suppressed. It can be seen from Figure 18a that the guidance line heading angle data jitter is obviously smoothed by the Kalman filter in section A, and even if the guidance line heading angle jitters frequently, the Kalman filter will make a smooth transition. Comparing Figure 18a,b, the Kalman filtering effect of the guidance lateral offset is better than that of the guidance heading angle. In section B, the abnormal data in the guidance lateral offset are obviously filtered, and the average error between the filtered data and the original data is 2.26 cm, indicating that the Kalman filtering prediction effect is relatively accurate, and the data after Kalman filtering can be sent to the row-oriented spraying system as a control signal.

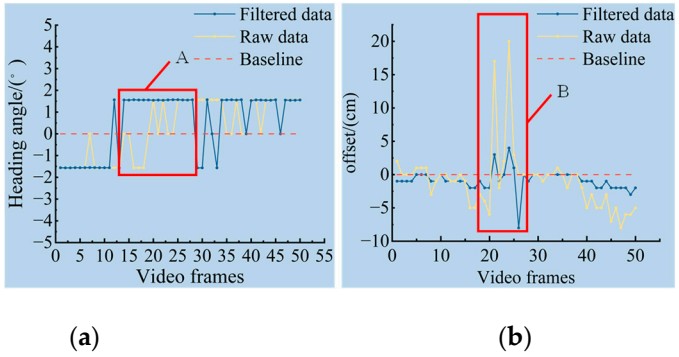

        **(a)**                                **(b)**

**Figure 18.** (**a**) Diagram of heading angle; (**b**) Diagram of offset.

*3.4. Results and Analysis of the Row Alignment Test*

As shown in Figure 19, in three different placement conditions, the row alignment track can follow the baseline. If placed in parallel, the average error of the alignment offset is 3.2 cm, as shown in Figure 19a, and some of the following tracks in section A deviate from the baseline. Figure 19e also indicates that the lodging of crop leaves changes the position of feature points obtained using the row guidance algorithm, resulting in a decrease in guidance line precision and an increase in the average error of the row offset. If placed obliquely, the average error of alignment deviation is 7.6 cm. Figure 19b shows that the inclination angle of placing crop rows changes slightly, resulting in a small change in the alignment offset provided by the algorithm, which increases the cumulative error of the row alignment mechanism in the moving process. Therefore, the average error of alignment deviation of oblique placement is the largest group among the three placement methods. In the test of the "S"-shaped display, the average deviation error is 5 cm, and the row alignment tracking patch in section B of Figure 19c correspond to the sand mark in Figure 19f, showing that the guidance algorithm can identify crop rows with great changes.

Generally, in the three placement conditions, the control system can adjust and control the row-oriented mechanism according to the row-orientation delay compensation model.

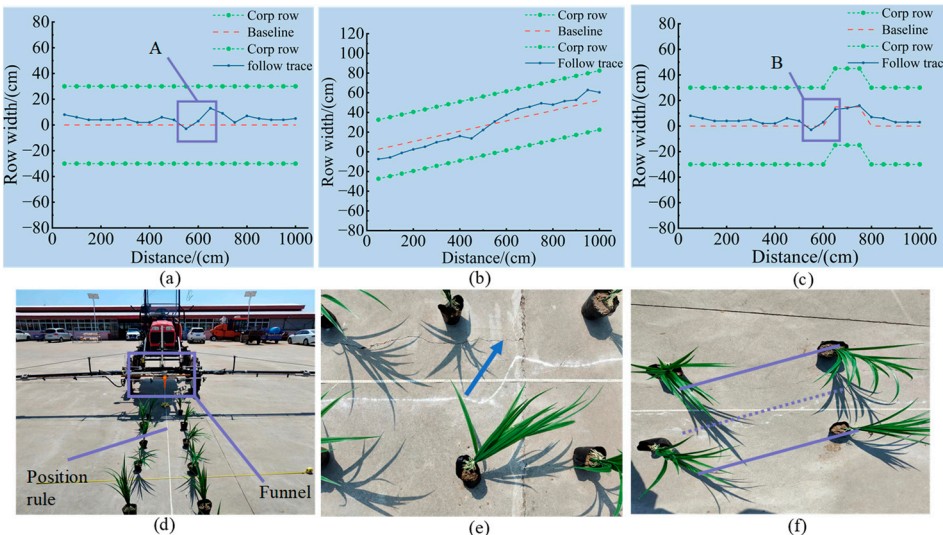

**Figure 19.** Row alignment test. (**a**) Parallel test; (**b**) Inclination test; (**c**) S-shaped test; (**d**) Testing layout; (**e**) Section A; (**f**) Section B.

### 3.5. Field Test

The test results are shown in Table 3. The test data show that the visual deviation of the guidance line and the row-oriented deviation of the row alignment control system gradually increased with the vehicle speed. When the speed was 0.27 m/s, 0.51 m/s and 0.67 m/s, the proportion of row-oriented deviation within ±15 cm was 86.66%, 90% and 90%, respectively. By observing the standard deviation of the row-orientation deviation in Table 3, the standard deviation when automatic navigation was turned on is obviously lower than that when automatic navigation was turned off, because when automatic navigation was turned off, the driver of the vehicle needed to adjust the steering wheel frequently, and with the increase in the speed of the test platform, the adjustment angle range of the steering wheel was enlarged, causing the chassis traveling track to fluctuate greatly, affecting the identification and row alignment control results. Furthermore, when automatic navigation was turned on, the row-orientation deviation samples were distributed more evenly, and the row-orientation effect was more stable. When the automatic navigation was turned on, with the increase in the speed of the test platform, the proportion of the row-orientation deviation within ±15 cm was 100%. When the automatic navigation was turned on, when the speed of the test platform was 0.71 m/s, the maximum deviation of the row orientation was 7.24 cm, meeting the requirements of automatic row alignment.

**Table 3.** Field tests.

| Navigation State | Speed m/s | Visual Mean Deviation/cm | Visual Standard Deviation | Row-Orientation Mean Deviation o/cm | Row-Orientation Standard Deviation/cm | Proportion of Row-Orientation Deviation within ±15 cm/% | Proportion of Row-Orientation Deviation within ±30 cm/% |
|---|---|---|---|---|---|---|---|
| off | 0.27 | 0.37 | 2.78 | 5.22 | 5.08 | 90 | 100 |
| | 0.51 | 0.14 | 2.01 | 7.92 | 7.98 | 90 | 100 |
| | 0.67 | 0.72 | 5.84 | 8.43 | 8.75 | 86.66 | 100 |
| on | 0.32 | 0.31 | 1.06 | 3.7 | 3.08 | 100 | 100 |
| | 0.56 | 0.46 | 1.23 | 4.36 | 4.54 | 100 | 100 |
| | 0.71 | 0.45 | 3.60 | 7.24 | 7.64 | 100 | 100 |

### 3.6. Spraying Performance Test

After the droplet deposition was obtained, the droplet deposition per unit area was normalized using the Origin drawing software, and the heatmaps of the row-oriented spraying and full-coverage spraying were drawn. Figure 20a shows that there was a significant difference in the color depth of droplet deposition between rows and between plants, but the inter-row color depth with row-oriented spraying was very close to the full-coverage spraying interrow color depth in Figure 20b, indicating that the deposition between them is also close. The two pictures show that the uniformity of the full-coverage spraying is better than that of the row-oriented spraying. In addition, the uniformity of the row-oriented spraying in the row is better than that between plant depositions. From the color change of heatmap 20(a), it can be inferred that the droplet deposition of the row-oriented spraying is mainly distributed on plants.

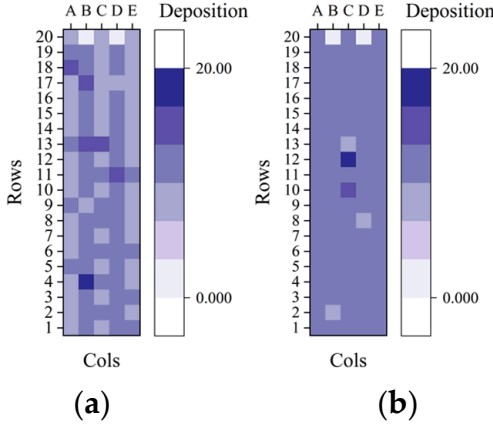

**Figure 20.** (**a**) Deposition of row-oriented spraying; (**b**) Deposition of full-coverage spraying.

The droplet deposition of each group was summarized, and data normalization and mean processing were performed for the droplet deposition data of row-oriented spraying and full-coverage spraying. The test results shown in Table 4 indicate that the droplet deposition of row-oriented spraying in columns A, C and E decreased by 18.22%, 18.81% and 21.01%, respectively, compared with row space application of B and D. However, the deposition of A, C, E of full coverage spraying decreased by 19.28%, 19.86% and 22.03%, respectively, compared with row-oriented spraying. According to the statistics of the total deposition of the two spraying methods, the total deposition of full-coverage spraying is 1039 mL, and 920 mL for row-oriented spraying. Comparing the two methods, the overall pesticide saving rate is 11.4%. According to the statistics of the inter-row deposition of the two spraying methods, the inter-row deposition of full-coverage spraying is 647 mL, and that of row-oriented spraying is 515 mL. The inter-row pesticide saving rate is 20.4%, which indicates that the row-oriented pesticide application scheme can reduce inter-row pesticide loss.

**Table 4.** Deposition statistics.

| Deposition mL/m$^2$ | | | | | |
|---|---|---|---|---|---|
| Col | A | B | C | D | E |
| Full-coverage spraying | 11.49 | 11.43 | 12.31 | 11.46 | 12.15 |
| Row-oriented spraying | 9.67 | 12.11 | 9.60 | 11.54 | 9.34 |

## 4. Discussion

In this paper, seedling stage maize was taken as a research object, focusing on the operation effect and spraying performance of a visual row-oriented control system. Considering the timeliness of information perception, real-time extraction of guidance lines was realized.

Only when the algorithm takes at least 67 ms is the identification relatively smooth [36]. A real-time row guidance algorithm that takes 42 ms was designed. Between a traditional Hough transform [22] and Wang's rice seedling row detection based on a feature point neighborhood Hough transform [37], the former has obvious advantages. In row guidance research, experts pay more attention to the research on crop row extraction algorithms, and the guidance information is not determined after crop row extraction, so the perception layer cannot provide information decisions for the control system [38]. Therefore, an outlier filtering algorithm based on the slope characteristics of crop rows to obtain two auxiliary lines of crop rows is proposed, which adopts a tangent formula of the included angle to fit the guidance line. It can reduce redundant information and provide accurate visual data for the row alignment system. Regarding the complex interference factors in field operations, the guidance line "jitter" problem in continuous video frames was solved to improve the stability of visual processing. A Kalman filter was embedded in the visual algorithm to optimize the estimation of the guidance line. This was compared with a method of triggering photos with a camera interval of 300 ms [39], which will not cause a loss of guidance information due to a sudden change in the acquisition environment.

When working in the field, the row-orientation tracking effect requires the control system to be very precise. With increasing vehicle speed, the stability of manual alignment decreases. As shown in Table 3, when the vehicle speed was 0.67 m/s, the standard deviation of the visual deviation was 5.84 cm, and when the vehicle speed was 0.71 m/s, the standard deviation of the visual deviation was 3.60 cm when the navigation system was turned on. The autonomous navigation system was used for preliminary row alignment, which reduced the influence of chassis vibration on the visual extraction effect and further proved that the alignment accuracy can be improved by combining the navigation system with the row-oriented control system. The guidance system uses machine vision as the information perception scheme. After obtaining the guidance information, the control system obtained the actual coordinates of the aiming point through visual ranging, which improves recognition accuracy. Under the condition that automatic navigation is enabled and the speed is 0.71 m/s, the row-oriented deviation of the system is within the range of $\pm15$ cm. Compared with the Zhang [21] method of using laser radar to initiate target matching and mechanical sensor feedback logic signal for crop rows, the visual scheme effect is better. On the premise of ensuring accurate row alignment, a comparative test of full-coverage spraying and row-oriented spraying was designed to explore the droplet deposition of the two spray methods. As shown in Figure 20a, abnormal droplet deposition in the row-oriented spray mostly occurs between rows, which is caused by the movement of the sliding boomer and the drift of droplets with inertia. In the future, the stability of the row-orientation mechanism should be adjusted and the drift-proof nozzle should be replaced to reduce the deposition of pesticides between rows and improve the accuracy of row-oriented spraying.

## 5. Conclusions

This study explores the possibility of a visual row-orientation spraying system based on autonomous navigation. The following is a summary of the main research work presented in this paper.

1.  Because of the pesticide waste and environmental pollution caused by continuous spraying operations at the maize seedling stage, a set of visual row-oriented spraying systems based on automatic navigation technology was developed. Automatic navigation was conducted for preliminary row alignment, and machine vision technology was used to achieve accurate row alignment, which provided a new idea for fine plant protection operations.
2.  A Hough transform algorithm was used to detect the crop rows to be selected, an outlier threshold was eliminated based on the slope of the line to fit the auxiliary lines, and a guidance line was fitted according to the tangent formula. Because of the guidance line jitter problem of the extraction algorithm in video reasoning, a Kalman

filter was used to track the target of the guidance line, and a robust row guidance algorithm was designed. According to the requirements of visual localization and the parameters of various identifications, theoretical analysis was carried out to set up a row-orientation delay compensation model, and a row-orientation control algorithm based on vision was designed.

3. Test results based on the visual row-oriented spraying system show that the average time of the row guidance algorithm is 42 ms, and the average visual deviation is 2.75 cm, which can provide row guidance for the row alignment system. If automatic navigation is turned on, the row-orientation average deviation between the row alignment mechanism and the centerline between rows of crops is 5.08 cm, and the row-orientation deviation range is below ±15 cm. Compared with previous reports, the performance is improved, meeting the requirements of automatic row orientation. In addition, compared with traditional spraying, the inter-row deposition of visual row-oriented spraying is reduced by 20.36%, and the overall pesticide savings are 11.4%, which reduces pesticide waste and improves pesticide utilization. With increasing vehicle speed, the alignment accuracy also decreases. To overcome the limitation of speed on the effect of alignment, we will start by improving the visual algorithm efficiency and optimizing the hardware system structure and continue to try more innovative control schemes to improve the performance of the alignment operation.

**Author Contributions:** Conceptualization, X.Z. and K.Z.; Methodology, K.Z., C.Z. (Changyuan Zhai), C.H. and X.Z.; Validation, Y.H.; Formal analyses, C.H. and Y.H.; Investigation, K.Z. and X.Z.; Resources, C.Z. (Changyuan Zhai), C.Z. (Chunjiang Zhao); Data curation, K.Z.; Writing—original draft, K.Z.; Writing—review and editing, X.Z. and C.Z. (Changyuan Zhai); Funding acquisition, X.Z. and C.Z. (Changyuan Zhai); Supervision, X.Z., C.Z. (Changyuan Zhai) and C.Z. (Chunjiang Zhao). All authors have read and agreed to the published version of the manuscript.

**Funding:** Support was provided by (1) Special project of strategic leading science and technology of Chinese Academy of Sciences (XDA28090108); (2) Jiangsu Province and Education Ministry Co-sponsored Synergistic Innovation Center of Modern Agricultural Equipment (XTCX1002); (3) National Natural Foundation of China (32201647); (4) National Key R&D Program Project (2022YFD2001402); (5) Postgraduate Research Innovation Program of Xinjiang Agricultural University (XJAUGRI2022002).

**Institutional Review Board Statement:** Not applicable.

**Data Availability Statement:** The data presented in this study are available on request from the corresponding author.

**Conflicts of Interest:** The authors declare no conflict of interest.

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
