# Peer review of "Design and Experiment of an Automatic Row-Oriented Spraying System Based on Machine Vision for Early-Stage Maize Corps"

_agriculture, doi:10.3390/agriculture13030691_

Round 1
Reviewer 1 Report
Dear Authors.
An article describes important issue of management of spraying and its accuracy pesticides as well as improving pesticide utilization. The topic is on the boarder between agriculture and science about sensors and image elaboration. The innovation in this article is spraying platform which was built based on autonomous navigation. Particularly important is the problem of finding a solution that is resistant to changing external factors. Some improvements are required in the text.
Anyway I have following remarks
1. Reformulating in sentence L. 98-100, Figure 1 doesn’t show the purpose of the article. Better place for it is methods part. The algorithm always consists of sequence of particular steps.
2. There is no need to divide Methods in subchapters with third level. Pointing like: 2.1, 2.2 etc. is enough- change if possible. Title of subchapter 2.2.3. requires necessary supplement - pretreatment of what?
3. Discussion part should be rewritten, avoiding unnecessary repeats like ”this study shows, presents … etc.”
Author Response
Response to Reviewer 1 Comments
Article Title: “Design and Experiment of an Automatic Row-oriented Spraying System Based on Machine Vision for Early-Stage Maize Corps”
Dear Reviewers,
On behalf of all the authors, we sincerely appreciate your valuable comments on the manuscript. Your comments not only provided constructive suggestions for improving the quality of the manuscript but also led us to consider our approaches and the design of the system in detail. These comments will also promote our future research.
Best regards,
Kang Zheng , Xueguan Zhao, Changyuan Zhai *,Chunjiang Zhao *
Point 1: Reformulating in sentence L. 98-100, Figure 1 doesn’t show the purpose of the article. Better place for it is methods part. The algorithm always consists of sequence of particular steps.
Response 1: According to the suggestions of reviewers 1 and 2, readjust the text structure in article L.147-158, and highlight the research objectives and novelty in the introduction according to the existing problems in this research direction. In addition, Figure 1 is adjusted to method part L.427 for a more reasonable representation.
Point 2: There is no need to divide Methods in subchapters with third level. Pointing like: 2.1, 2.2 etc. is enough- change if possible. Title of subchapter 2.2.3. requires necessary supplement - pretreatment of what?
Response 2: We tried to shorten the sub-titles of the article and found that doing so lost the original logic of the article, so we merged some chapter and adjusted the sub-titles. Please see L.284, L.393, L.609, L.666 for details. Besides, the sub-titles supplement suggested by the experts have been revised, as shown in L.234.
Point 3: Discussion part should be rewritten, avoiding unnecessary repeats like “this study shows, presents … etc.”
Response 3: The discussion section has been rewritten to avoid the repeats mentioned in the review comments. Please see L.685-727 for details

Reviewer 2 Report
General comments:
The manuscript entitled " Design and Experiment of an Automatic Row-oriented Spraying System Based on Machine Vision for Early Stage Maize Corps" developed an automatic row-oriented control system based on a high clearance sprayer. The topic of the article is relevant and may be of interest to specialists and researchers in the fields of agriculture.
The research article is well written and contributes to the existing knowledge. The experiments appear to be well planned, results are interesting, the ideas and methods are correct. In my mind, the manuscript is acceptable for publication in Agriculture.
I suggest minor corrections to the authors, which are further listed:
Specific comments:
- The authors could clearly include more results and the impact of the research in the Abstract.
- Please split this paragraph (lines 48–79).
- Too many references from the same journal! Three updated references published in the Agriculture Journal can be added to the introduction.
- The novelty of the research must be highlighted in both the abstract and the introduction.
- The figures are informative, but the font sizes of the texts inside the figures (8, 10, 18, 19, and 20) should be increased for better presentation.
- Conclusions can be presented in points rather than paragraphs.
- Future plans and impacts of this research should be clearly and concisely mentioned at the end of the conclusion section.
Author Response
Response to Reviewer 2Comments
Article Title: “Design and Experiment of an Automatic Row-oriented Spraying System Based on Machine Vision for Early-Stage Maize Corps”
Dear Reviewers,
On behalf of all the authors, we sincerely appreciate your valuable comments on the manuscript. Your comments not only provided constructive suggestions for improving the quality of the manuscript but also led us to consider our approaches and the design of the system in detail. These comments will also promote our future research.
Best regards,
Kang Zheng , Xueguan Zhao, Changyuan Zhai *,Chunjiang Zhao *
Point 1: The authors could clearly include more results and the impact of the research in the Abstract.
Response 1: As suggested by reviewer 1.2, more visual row-oriented spraying results and research implications have been added to the abstract of the article. Please see L.17-34 for details.
Point 2: Please split this paragraph (lines 48–79).
Response 2: The paragraphs referred to by the reviewer have been separated, as detailed in L.67-96.
Point 3: Too many references from the same journal! Three updated references published in the Agriculture Journal can be added to the introduction.
Response 3: The references from agriculture and other journals have been added to the introduction to increase the diversity of references and backgrounds. Please see the references for details.
Point 4: The novelty of the research must be highlighted in both the abstract and the introduction.
Response 4: According to the suggestions of reviewers 1 and 2, readjust the article, and highlight the research objectives and novelty in the abstract and the introduction according to the existing problems in this research direction. Please see the abstract and the introduction for details.
Point 5: The figures are informative, but the font sizes of the texts inside the figures (8, 10, 18, 19, and 20) should be increased for better presentation.
Response 5: According to the review comments, we solved the font problems in Figures 8, 10, 18, 19, and 20.
Point 6: Conclusions can be presented in points rather than paragraphs.
Response : According to the review comments, we have formulated points in the conclusion, please see L771-804 for details.
Point 7: Future plans and impacts of this research should be clearly and concisely mentioned at the end of the conclusion section.
Response 7: We have added the future plans and impacts of the study to points 1 and 3 of the conclusion, please see L771-804 for details.
